# Metabolic Changes in Polycystic Kidney Disease as a Potential Target for Systemic Treatment

**DOI:** 10.3390/ijms21176093

**Published:** 2020-08-24

**Authors:** Sophie Haumann, Roman-Ulrich Müller, Max C. Liebau

**Affiliations:** 1Department of Pediatrics, University of Cologne, Faculty of Medicine and University Hospital Cologne, 50937 Cologne, Germany; sophie.haumann@uk-koeln.de; 2Department II of Internal Medicine, University of Cologne, Faculty of Medicine and University Hospital Cologne, 50937 Cologne, Germany; roman-ulrich.mueller@uk-koeln.de; 3CECAD, University of Cologne, Faculty of Medicine and University Hospital Cologne, 50931 Cologne, Germany; 4Systems Biology of Ageing Cologne, University of Cologne, 50931 Cologne, Germany; 5Center for Molecular Medicine, University of Cologne, Faculty of Medicine and University Hospital Cologne, 50931 Cologne, Germany

**Keywords:** ARPKD, ADPKD, cilia, polycystin, fibrocystin, cellular metabolism

## Abstract

Autosomal recessive and autosomal dominant polycystic kidney disease (ARPKD, ADPKD) are systemic disorders with pronounced hepatorenal phenotypes. While the main underlying genetic causes of both ARPKD and ADPKD have been well-known for years, the exact molecular mechanisms resulting in the observed clinical phenotypes in the different organs, remain incompletely understood. Recent research has identified cellular metabolic changes in PKD. These findings are of major relevance as there may be an immediate translation into clinical trials and potentially clinical practice. Here, we review important results in the field regarding metabolic changes in PKD and their modulation as a potential target of systemic treatment.

## 1. Introduction

The autosomal recessive and autosomal dominant forms of polycystic kidney disease (ARPKD, ADPKD) are among the most common causes of end stage renal disease in children and adults [1]. Over the last 20 years the pathophysiological understanding of ADPKD and ARPKD has gained increasing interest and the excellent work of multiple research groups worldwide has led to the identification of the underlying variants of different forms of cystic kidney diseases and of various intracellular signaling pathways that are dysregulated in PKD [2,3,4,5,6,7,8]. Based on these findings large clinical trials have been established resulting in the identification of the V2-receptor antagonist tolvaptan as a first targeted treatment of ADPKD [9,10,11]. Most recent cellular research has highlighted a potential role for cellular metabolic changes in the pathogenesis of PKD, thus opening a novel road for potential therapeutic approaches. Here, we give a brief overview over important recent developments in this area of research and the potential relevance for future therapeutic options.

## 2. Polycystic Kidney Disease

The main characteristics of the clinical course and the underlying genetics of ARPKD and ADPKD have recently been summarized in various excellent reviews [3,4,5,6,7,8]. Briefly, ARPKD is the severe form of polycystic kidney disease that frequently presents during early childhood or even antenatally. ARPKD is still associated with a substantial mortality that was estimated to be as high as up to 30–40% of the children although updated numbers are missing and are hard to be obtained. ARPKD is characterized by bilaterally enlarged kidneys due to the development of tubular dilatations or microcysts that mainly derive from the distal nephron [2]. Macrocysts may be seen and are assumed to be more frequent in the medulla [12]. Kidney function varies in ARPKD, but studies have found that about 50% of the patients require renal replacement therapy by the age of 20 years [13]. There is obligatory hepatic involvement with congenital hepatic fibrosis due to ductal plate malformation that can lead to portal hypertension with a need for liver transplantation. Even though ARPKD is a rare disease with an estimated incidence of 1:20,000, it is one of the main indications for combined liver and kidney transplantation during childhood [14]. Overall, there is a high degree of phenotypic variability that is poorly understood. Current approaches try to address this through international collection of clinical courses to lay the foundation for treatment recommendation based on observational evidence [15,16,17].

Most of the cases of ARPKD are caused by variants in the *PKHD1* gene, although patients with variants in additional genes (e.g., *DZIP1L* encoding DAZ interacting protein 1-like, *PMM2* encoding Phosphomannomutase 2) leading to an ARPKD phenotype have been described [6,18]. *PKHD1* encodes a 450 kDa protein with a single transmembrane domain called fibrocystin. Fibrocystin’s cellular function remains very poorly understood. The cytoplasmic tail of fibrocystin has been shown to contain a ciliary targeting signal [19]. Upon cleavage the cytoplasmic tail can translocate to the cell nucleus [20,21]. Yet the functional consequences of these observations and the pathophysiological relevance remain unclear [22,23]. Studies on fibrocystin’s cellular function have been hampered by the fact that orthologous mouse models do not fully recapitulate the phenotype of ARPKD patients [23,24,25,26].

ADPKD typically presents later in life than ARPKD and is the more common form of PKD. An estimated incidence of 1:400–1:1.000 has been suggested but underdiagnosis makes accurate estimations a challenge. Most of the patients become clinically apparent from the fourth decade of life and end-stage renal disease occurs in the sixth decade in about 50% of affected individuals. ADPKD is classically characterized by the progressive development of bilateral renal cysts resulting in kidney enlargement. Liver cysts are seen in the vast majority of patients. Other extrarenal manifestations may occur as recently reviewed [27,28]. Mortality is primarily driven by renal failure associated with cardiovascular complications and is estimated to be increased 1.6 times relative to the general population. Very early onset forms of ADPKD have been described that may clinically resemble ARPKD [29].

ADPKD is mainly caused by variants in the *PKD1* or the *PKD2* gene encoding the proteins polycystin-1 and polycystin-2. Most patients show variants in *PKD1* (around 85%) [30]. *PKD1* variants are associated with a more rapidly progressing phenotype than *PKD2* variants [31]. Importantly, truncating *PKD1* variants have to be distinguished from non-truncating changes since the latter come along with a much milder disease course [32]. Recently, variants in additional genes (*ALG9* encoding Alpha-1,2-mannosyltransferase ALG9, *GANAB* encoding glucosidase II subunit α (GIIα) and *DNAJB11* encoding DnaJ heat shock protein family (Hsp40) member B11) have been described to cause a phenotype resembling ADPKD in smaller subsets of patients [33,34]. First classifications have been established identifying ADPKD patients at risk of fast progression [35,36].

Much has been learned about the function of the polycystin proteins during the past 15 years. A detailed description of the insights lies beyond the scope of this article and interested readers are referred to excellent recently published reviews [7,37,38] Briefly, polycystin-1 is a large transmembrane protein with a long extracellular part, eleven transmembrane domains and a short cytoplasmic tail that can be cleaved. Polycystin-2 is a nonselective cation channel and belongs to the family of transient receptor potential ion channels (TRPs). Both proteins have been found to physically interact and to function in joint complex. However, there may also be independent roles for cellular signaling [39].

Patient data and data from animal models of ADPKD and ARPKD furthermore suggest that there may be partial overlap in the cellular pathogenesis of the different types of PKD. Patients with variants in multiple PKD genes have been described that showed a severe phenotype [29] and crossings of a *Pkd1* to a *Pkhd1* mouse model resulted in a more severe phenotype [24,40]. A link between Fibrocystin and Polycystin-2 has also been described [41]. Recently Lea et al. could show that Polycystin-1, Polycystin-2, and Fibrocystin are detectable as a polycystin complex (PCC) in human urinary exosome like vesicle, that all of them undergo post-translational proteolytic processing and that there are various cleavage events in all three proteins exceeding the well described ones [42].

Despite the enormous progress, the exact mechanisms resulting in the initial development of a cyst and the following expansion of a cyst remain incompletely understood. Cysts in ADPKD are considered to be the result of a clonal expansion of an affected cell [43]. In line with this hypothesis, a second (or even a third) hit has been reported to be required in the heterozygous renal epithelial cells to induce the process resulting in cystogenesis in ADPKD [44,45,46]. Multiple signaling pathways have been found to be dysregulated in ADPKD with increased intracellular cyclic adenosine monophosphate (cAMP) concentration potentially as a consequence of vasopressin V2-receptor activation being a very prominent example that resulted in the identification of tolvaptan as a first treatment option of ADPKD [9,10,11] Increased intracellular cAMP concentrations also result in activation of the extracellular-signal regulated kinase and mitogen-activated protein kinase pathway (ERK/MAPK pathway) [47,48].

## 3. Mammalian Target of Rapamycin (mTOR) Activation in PKD

An aspect that has recently received increasing interest in the field are the changes of cellular metabolism in PKD. Other proliferative disorders like different tumor entities have given substantial insights into changes of cellular signaling and cellular metabolism that may also be relevant in PKD [49].

One of the starting points for this research area was the detection of an activation of the mTOR pathway in murine models of PKD [50,51]. mTOR itself is an atypical serine/threonine kinase that can be found in mTOR Complex 1 (mTORC1) and mTOR Complex 2 (mTORC2) with independent cellular functions. mTOR signaling is involved in a multitude of cellular processes related to cell growth and metabolism [52,53,54]. mTORC1 activity can be regulated through multiple intra- and extracellular ways including the presence of growth factors, the availability of nutrients and the energy status of the cell and can be controlled by the drug rapamycin [53,55,56]. In brief, activation of mTORC1 at the lysosomal membrane is regulated by sensing of intracellular nutrients and various upstream pathways resulting in control of Rag GTPases and members of the *Ragulator* complex and thus mTORC1. Activation of mTORC1 results in promotion of anabolic processes and the inhibition of catabolism including autophagy [57].

Activation of the mTOR pathway was observed in cyst lining epithelia of ADPKD kidneys and polycystin-1 was shown to negatively regulate mTOR activation [51]. Inhibition of mTOR using rapamycin reduced kidney volume in different animal models and resulted in better renal function [50]. These observations led to the establishment of large clinical trials for the use of mTOR inhibitors in ADPKD. Yet, these trials failed to show the hoped-for breakthrough results. In the study by Walz et al. 433 patients with advanced ADPKD in chronic kidney disease (CKD) stage G3a were treated with everolimus or placebo in a 2-year, randomized, double-blinded, placebo-controlled trial [58]. Based on the findings by the CRISP consortium and others total kidney volume determined by magnetic resonance imaging (MRI) was used as a surrogate for disease progression [59,60,61,62]. The trial could show a significant reduction in increase of total kidney volume in the everolimus group after one year, but this effect lost statistical significance after the second year. Importantly, decline of the estimated glomerular filtration rate (eGFR) was more rapid in patients on everolimus [58]. The SUISSE ADPKD trial studied 100 patients with an eGFR above 70 mL/min in an open-label randomized, controlled trial using sirolimus vs. placebo over 18 months. There were no significant differences in total kidney volume or eGFR between the groups but sirolimus patients showed more albuminuria, a known side effect of this drug [63,64].

It is important to point out that the therapeutic dosage used in animal models could not be achieved in clinical trials and that the animal models initially used for evaluation of mTOR inhibition do not fully recapitulate ADPKD but rather were models of rapidly progressing PKD [50,65,66,67]. Furthermore, the dropout-rate in the treatment arm was substantial in the everolimus study and the design of the clinical trials may not have detected all positive effects of mTOR inhibition on ADPKD progression in tubular cells [64,68,69,70]. It has been suggested that a direct delivery of mTOR inhibitors to renal tubular epithelial cells may have more specific positive effects. Indeed, modified rapamycin that was conjugated to folate with the aim to specifically target proliferating cells in cystic epithelium did show positive effects on cyst growth and kidney function in rapidly progressing PKD mouse model [71].

Multiple ways of interaction between the ADPKD protein polycystin 1 (PC1) and mTOR signaling have been described [72]. PC1 overexpression leads to a downregulation of the MEK/ERK pathway resulting in decreased activation of tuberin and thus mTOR inhibition. PC1-deficiency on the other hand results in an upregulation of ERK1/2 and through this to an upregulation of the mTOR pathway [73]. Vice versa the activation of mTORC1 in mouse embryonic fibroblasts (MEFs) was shown to have a negative influence on the expression of PC1, as well as on the trafficking of PC1/PC2 complex into cilia [74]. An effect of fibrocystin on mammalian target of rapamycin (mTOR) activation has been found in vitro [75] and activation of mTOR signaling has been described in cyst lining epithelium in vivo [76]. Treatment with NVP-BEZ235, an inhibitor of PI3-kinase and mTOR, had a positive effect on cystic dilatation of the intrahepatic bile ducts in the PCK rat [77]. 

In a *pkd1*-deficient zebrafish model has provided an insight into mechanisms downstream of mTOR activation that have an impact in PKD. Inhibition of autophagy by knocking down core autophagy proteins like Afg5 let to increased cystogenesis while activation of autophagy via Beclin-1 peptide ameliorated cystogenesis. These mechanisms seem to be conserved in mouse models and human tissue [78]. Furthermore, the treatment with low dose rapamycin and carbamazepine, known to activate autophagy in a mTOR independent way, was very effective in zebrafish [78] pointing towards a potential role of combination therapies that tackle this pathway at different levels.

## 4. Glucose Metabolism, Glycolysis, Oxidative Phosphorylation, and the Warburg Effect

A novel link of cellular metabolism to ADPKD has more recently emerged pointing to an important role of cellular glucose metabolism in ADPKD [79]. In healthy cells, adenosine triphosphate (ATP) production is known to take place in the mitochondria in the presence of oxygen. For this, pyruvate emerging from glycolysis is metabolized to acetyl coenzyme A (Acetyl-CoA) through oxidative decarboxylation. Acetyl-CoA, which is the main product entering the Krebs cycle is processed in the mitochondrial matrix resulting in the generation of nicotinamide adenine dinucleotide (NADH) and flavin adenine dinucleotide (FADH2). NADH and FADH2 then enter the oxidative phosphorylation cascade via complex I and complex II of the respiratory chain. Via electron changes a transmembrane gradient between inner and outer mitochondrial membrane is built up and enables the flux of protons which is finally used to regenerate ATP from adenosine diphosphate (ADP) by the use of oxygen at complex V. This process called oxidative phosphorylation is highly effective and produces 36 molecules ATP out of one molecule glucose. In the absence of oxygen, pyruvate is converted into lactate in an enzymatic reaction in the cytosol through lactate dehydrogenase leading to the production of two molecules ATP and regeneration of two molecules NADH [80].

In 1924, Otto Warburg made the observation that tumor cells use the metabolic conversion of pyruvate into lactate even in the presence of oxygen. This so-called aerobic glycolysis [81], has been a starting point to many newly discovered pathways in the last years, summarized in various excellent reviews [82,83,84,85]. Reprogramming of energy metabolism has been added to one of the emerging hallmarks for tumorigenesis in 2011, emphasizing the importance of changes in cancer cell metabolism [86].

By metabolomic characterization *Pkd1*-deficient murine cells were recently found to also preferentially use the approximate 18-fold less effective aerobic glycolysis even in the presence of oxygen. Tumor cells upregulate multiple genes involved in glucose transport and glycolysis [87,88] and indeed, this was found to also be the case in *Pkd1*-deficient cells [88]. This finding was confirmed in human tissue [89,90]. Importantly, treatment of two different *Pkd1*-mouse models of ADPKD with 2-deoxyglucose, an inhibitor of glycolysis showed beneficial results with lower kidney weight and volume and normalized AMP-activated protein kinase and its target acetyl-CoA carboxylase [89,91]. The treatment with 2-deoxyglucose was also successful in a rat PKD model [92].

In addition, there were first hints that mitochondrial protein levels as well as the activity of crucial subcomplexes of the mitochondrial electron chain were altered in a Mini-pig PKD model [93]. Most recently it was found that a cleavage product of polycystin-1 can affect mitochondria morphology and function and that mitochondria of ADPKD patients show structural changes [94,95]. Besides, PC1 and PC2 localize in mitochondria associated membranes and can interact with the calcium signaling from the lumen of the ER to the mitochondrial matrix [96,97,98].

Cassina et al. could show that increased mitochondrial fragmentation seems to be a modifier of disease progression. In *Pkd1* mutant mice the authors could show aberrant mitochondria in the cyst-lining epithelial cells with a reduced mitochondrial mass and a following reduction in mitochondrial respiratory chain complexes. This seems to be due to a fragmentation of the mitochondrial network which is driven by a downregulation of the pro-fusion proteins OPA-1 (mitochondrial Dynamin-like 120 kDa protein/optic atrophy protein 1) and MFN-1 (Mitofusin-1) and an increase in the pro-fission protein DRP-1 (Dynamin-related protein 1) [99].

## 5. Amino Acid Metabolism

PKD cells, very much like cancer cells, seem to have a high need in biomass while switching into an anabolic mode [83]. One of the essential carbon sources in this context is glutamine, which can be catabolized into α-ketoglutarate (α-KG) through a process called glutamine anaplerosis. Glutamine is converted into glutamate and glutamate into α-KG, which itself is enters the TCA, fueling the electron transport chain and maintaining the mitochondrial membrane potential. [100]

For PKD, glutamine addiction was first described by Hwang et al. [101]. A subsequent study showed that glutamine-dependence caused by ablation of *Lkb1* in combination with ablation of *Tsc1* gene resulted in very aggressive cystogenesis in a mouse model and inhibition of glutamine metabolism in both *Lkb1*/*Tsc1* and *Pkd1* mutant mice significantly reduced cyst progression [102].

In 2018, Podrini et al. were able to show that indeed glutamine usage is upregulated in PKD, most likely to keep up the mitochondrial membrane potential and to deliver enough building blocks for rapid cell division. Furthermore, they showed that glutamine is converted into glutamate via asparagine synthetase (ASNS), a rather unusual mechanism. ASNS converts aspartate into asparaginase while converting glutamine into glutamate [103]. Beyond this they could show that in PKD cells glutamine-derived α-KG not only seems to be used by the cell to maintain the OXPHOS, but it is also carboxylated to produce citrate, which itself is converted into Actely-CoA and Oxalacetate (OAA). Actyl-CoA is used for the fatty acid synthesis and OAA is converted into aspartate, fueling the ASNS for an efficient transport of glutamine into the cells and the OXPHOS, closing the circle and underlining the glutamine dependence of PKD cell metabolism [96,103].

Alternative amino acid metabolism pathways have also been described for ADPKD cells. Trott et al. could show in 2018 that arginine regulates the enzyme arginosuccinate synthase 1 (ASS1) and is essential for cystogenesis in an ex vivo model (Figure 1) [104].

## 6. ERK-AMPK

AMP-activated protein kinase (AMPK) is a highly conserved eukaryotic kinase, which plays a crucial role in controlling cellular energy balance [105]. AMPK is activated by increases in AMP/ATP and ADP/ATP ratios with the result to increase cellular catabolism and to decrease cellular anabolic processes [106,107]. This includes stimulation of fatty acid oxidation and glucose uptake. AMPK leads to phosphorylation of raptor and tuberous sclerosis complex 2, which itself leads to inhibition of mTOR and elongation factor 2 ending in a decrease of protein synthesis, cell growth and cell proliferation [108]. Interestingly enough, AMPK leads to gluconeogenesis via inhibition of the transcription factor hepatocyte nuclear factor 4 (Hnf4) and inhibits mTORC1 [78,108]

In *Pkd1*-/-cells, the activating AMPK phosphorylation is decreased compared to wild-type cells [89]. As AMPK activation leads to mTOR inhibition, which has been described earlier in this manuscript as a potential strategy to decrease cyst growth in mouse model, the activation of AMPK pathway may be a good target for therapeutic strategies. AMPK itself was found to be regulated through an ERK1/2-Liver kinase B1 (LKB1) axis [89].The activation of the ERK1/2 pathway could be confirmed in a PKD mini-pig model [93].

AMPK activation may become an interesting therapeutic option as metformin is a drug with a broad clinical use in Type 2 diabetes mellitus and the advent of a first application in cancer treatment in the last decade [109,110]. Metformin is known to interfere with mitochondrial phosphorylation. The inhibition of complex 1 (NADH dehydrogenase) of the electron chain of oxidative phosphorylation could be shown in tumor cells making Metformin interesting for cancer treatment [111]. This inhibition of mitochondrial complex I was also shown in peripheral blood mononuclear cells and platelets [112], but has not been studied in detail in renal cells so far. Yet, metformin has been shown to activate AMPK in an ADPKD mouse model resulting in decreased cyst growth via inhibition of mTOR in vitro [113]. Moreover, it was shown that treatment of ADPKD murine embryonic fibroblasts with Metformin and other AMPK activators led to a decrease of the aerobic glycolysis or Warburg effect [89]. These findings could be confirmed in a PKD rat model [92]. An additional study found additive effects of rapamycin and metformin in counteracting mTOR activation after knockdown of *Pkd1* [114]. However, another study using a *Pkd1* knockout mouse model did not find beneficial effects of metformin [115]. To clarify potential efficacy in humans, a first clinical trial has been repurposing metformin for ADPKD (Metformin as a Novel Therapy for Autosomal Dominant Polycystic Kidney Disease (TAME), ClinalTrials.gov, Identifier: NCT02656017).

In a transcriptomic and metabolomic analysis of an early-onset murine model of PKD due to an induced knockout of *Pkd1* the transcription factor Hnf4α (Hepatocyte nuclear factor 4-alpha) was identified as a potential important regulator of metabolism affected in APDKD [116]. Hnf4α is a downstream target of AMPK [117] that is involved in the regulation of glucose metabolism [118,119,120]. Indeed, multiple genes involved in glucose regulation have been found to be regulated by SREBP1, HIF1-a or HNF4α in an mTOR-dependent way [121]. Interestingly, the double knockout of *Hnf4α* and *Pkd1* resulted in a more severe phenotype [116]. The authors speculated that the upregulation of Hnf4α might be a secondary compensatory mechanism (Figure 2) [116].

## 7. Fatty Acid Oxidation

Various links were established between fatty acid oxidation and PKD including a cystic renal phenotype in some disorders of fatty acid oxidation [122,123]. Indeed, it may be clinically challenging to differentiate these disorders [124]. Interestingly, overlapping metabolic features have also been found in disorders of fatty acid oxidation and PKD including activation of the ERK-mTOR axis, reduced oxidative phosphorylation and increased generation of lactate [124]. In addition, impaired fatty acid oxidation has been found in animal models of PKD [125] and an attenuation of both the renal and the hepatic phenotype has been described in a slowly progressing *Pkd1* mouse model after activation of fatty acid oxidation using the PPAR-α agonist fenofibrate [126]. This is in accordance with findings from the *PCK* rat, an orthologous model of ARPKD with a renal phenotype with features also resembling ADPKD [127].

## 8. Sirtuin

Sirtuin-1 (SIRT-1) is an NAD-dependent protein deacetylase and a known regulatory protein in apoptosis. Sirtuin-1 leads to a destabilization of p53, one of the best characterized pro-apoptotic factors, which is in mutated status relevant for tumor progression in many characterized diseases such as tumors in the adrenal gland [128].

In 2013 it was shown that *Sirt1* expression is upregulated in *Pkd1*-deficient cells and that the distal tubular double knockout of *Pkd1* and *Sirt1* leads to a less severe phenotype than the single knockout of distal tubular *Pkd1* in mouse model. Furthermore, the treatment of *Pkd1* lacking embryos in mouse model with the Sirtuin-1 inhibitor EX-527 or nicotinamide was leading to slower cyst growth in these animals [129]. Interestingly, SIRT-1 is a crucial metabolic NAD+-sensor. The presence of SIRT-1 leads to changes in the chromatin structure important for the expression of genes relevant for metabolism in different organs [130]. In case of high glycolytic levels in the cell it is crucial for them to regenerate NAD+ in the cytoplasm, which sums up in the conversion from pyruvate into lactate [131]. The role of this mechanism in PKD remains to be established.

## 9. Cilia, mTOR, and Cellular Metabolism

In addition to the link of polycystin-1 to mTOR signaling there may be a direct effect of cilia on mTOR and cellular metabolism. The link between cilia and PKD has been reviewed and is beyond the scope of this manuscript [132,133]. Yet, it has also been shown that ablation of cilia in a rodent model leads to enlarged cell size and that this process is driven by mTORC1 [134]. Ciliary bending seems to be important for mTORC1 inhibition and cell-size control. Cultivating ciliated cells under flow leads to decreased mTORC1 levels and via this to decreased cell size. Furthermore LKB-1, a tumor-suppressor expressed in cilia, drives AMPK-phosphorylation at the ciliary basal body and leads to further mTORC1 inhibition [134]. These alterations in the AMPK concentrations at the ciliary basal body may point to an important interplay of ciliary and metabolic aspects in the pathogenesis in PKD.

## 10. Emerging Treatment Options

The first two major clinical trials in the PKD field that targeted cellular metabolic aspects were the previously described trials on mTOR inhibitor use in ADPKD [59,65]. The insights obtained in basic science that we describe in this manuscript have a great additional potential to be directly translated into clinical life. Multiple clinical trials will be required until everyday clinical practice in the nephrology ward will directly be influenced by these insights. Given the recent developments, novel initiatives have been launched and novel potential treatment approaches are emerging. We have already described some of these options in the subsections including modified mTOR inhibitors, inhibition of glycolysis, activation of AMPK through Metformin, PPAR-α activation or Sirtuin inhibition. While most of the work has focused on the renal phenotype, there are also hints that extrarenal manifestations like the hepatic phenotype can be influenced. A detailed analysis of the effects on extrarenal manifestations will be required.

A very interesting and easy to access therapeutic strategy to reduce renal cyst growth in PKD via mild to moderate food or caloric restriction has recently been shown to work for different orthologous mouse models of ADPKD [135,136]. The reduction of caloric or food intake has been shown to be highly effective to increase longevity [137,138]. The underlying cellular mechanisms seem to involve a complex interplay of multiple signaling cascade including mTOR, Sirtuins, and AMPK and result in mTOR inhibition and AMPK activation, thus exactly the conditions that seem to be favorable in PKD. Indeed, in both publications a rather mild restriction of food resulted in positive effects on kidney volume. Upregulation of AMPK and/or downregulation of the mTOR pathway were shown in PKD animals undergoing mild to moderate caloric restriction [135,136]. Importantly, in humans, obesity is known to be a risk factor for rapid disease progression in ADPKD [139]. Lately Torres et al. could show that the benefit of mild reduction in food intake is driven by induction of ketosis in preclinical PKD models. Time-restricted feeding was shown to inhibit mTOR signaling, proliferation and fibrosis in rodent kidneys. The same effect was seen in ketogenic diet, during acute fasting and with administration of β-hydroxybutyrate (BHB) to normal chow in preclinical mouse, rat and cat models of PKD, thus widening the therapeutic options to fasting and nutritive supplementation of ketones [140]. A clinical pilot trial that will test potential acute effects on kidney volume—as observed in the feline model—is currently starting recruitment (NCT04472624) and a larger trial using a 3-month intervention is going to examine feasibility of ketogenic interventions in ADPKD (https://pkdcure.org/funded-research/). Considering the lack of data on dietary measures to improve outcome in ADPKD, more clinical trials are urgently needed to rapidly evaluate these widely-available and cost-effective therapeutic strategies.

## 11. Conclusions

In summary, recent work has identified multiple metabolic cellular changes in cyst-lining epithelia in PKD both in animal models as well as in patients. This emerging field is currently at the verge of being transferred into clinical life with the first ongoing clinical trials.

## Figures and Tables

**Figure 1 ijms-21-06093-f001:**
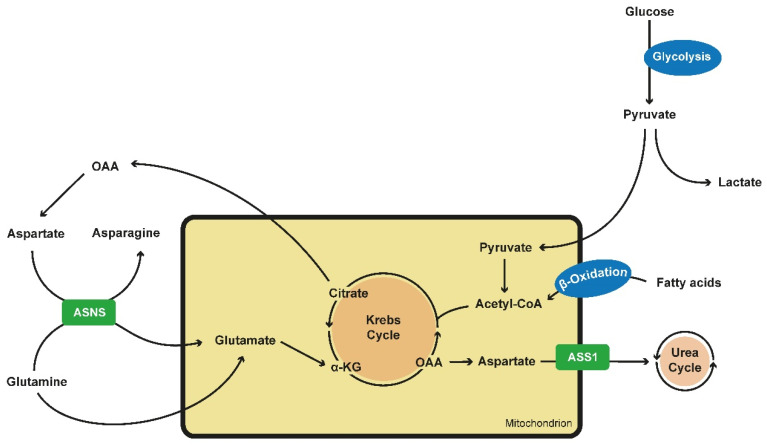
Alterations in metabolics in PKD cells. Glucose is preferably converted into lactate and pyruvate via aerobic glycolysis. As pyruvate is an inefficient fueling for the TCA or Krebs cycle, PKD cells tent to generate building blocks via the use of fatty acids via β-oxidation and the use of the glutamine metabolism. Glutamine is transported into the mitochondrion via ASNS, an enzyme which converts aspartate into asparagine while glutamine is converted into glutamate which itself is located into the mitochondrium. Here it can be fully oxidated and maintain the TCA cycle or it can be carboxylated to produce citrate. Citrate is converted into OAA and acetyl-CoA. OAA itself can generate aspartate, fueling ASNS. ASS1 converts aspartate into arginosuccinate, fueling the urea cycle and being the crucial step of de novo arginine synthesis. ASS1 expression is reduced in PKD cells. (PKD—polycystic kidney disease; TCA—tricarboxylic acid cycle; ASNS—asparagine synthethase; OAA—oxalacetate; ASS1—arginosuccinate synthase 1).

**Figure 2 ijms-21-06093-f002:**
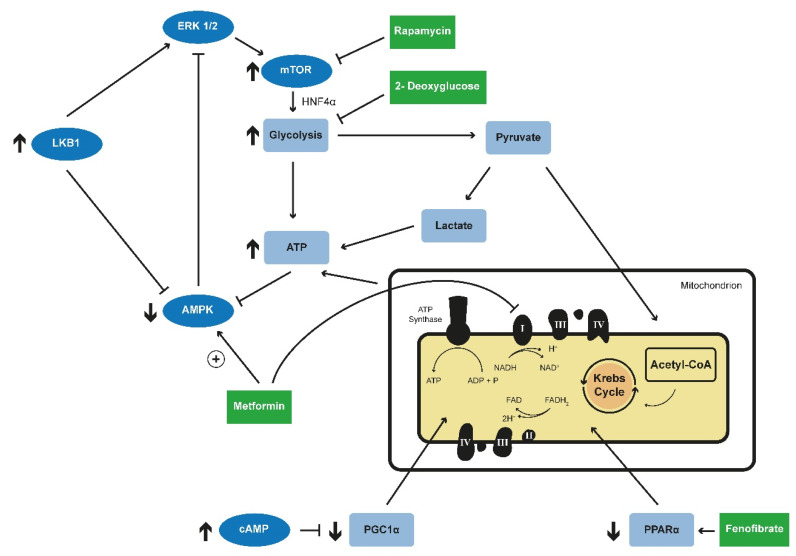
Targeted metabolic reprogramming in PKD. Deficiency of PC1 was shown to lead to an upregulation of ERK1/2, which further leads to an activation of the mTOR complex. Inhibition of mTOR by rapamycin led to promising results in animal models. Activating AMPK phosphorylation is decreased in *Pkd1*-lacking cells. AMPK leads to inhibition of mTOR and is regulated itself through an ERK1/2-liver kinase B1 (LKB1). Metformin, a well-known drug in the treatment of diabetes type 2, was shown to activate AMPK in an ADPKD mouse model resulting in decreased cyst growth via the inhibition of mTOR. Furthermore, treatment with Metformin reduced aerobic glycolysis in animal model. Metformin is also known to inhibit complex 1 of the mitochondrial electron chain. 2-Deoxyglucose is an inhibitor of glycolysis and the treatment with 2-Deoxyglucose in different animal models resulted in lower kidney weight and volume and normalized AMPK phosphorylation. HNF4α is a downstream target of AMPK and is involved in glucose metabolism. The double knockout of *Hnf4α* and *Pkd1* resulted in a more severe phenotype in animal model. Impaired fatty acid oxidation has been observed in PKD animal models and the reactivation of this process via the PPAR-α agonist fenofibrate led to an attenuation of the renal and hepatic phenotype. Increased intracellular levels of cAMP as a consequence of vasopressin V2-receptor activation are described in ADPKD. PGC1α induces and coordinates gene expression that stimulates mitochondrial oxidative metabolism. (PC1-Polycystin1; mTOR—mammalian target of rapamycine; AMPK—adenosine monophospahte-activated protein kinase; HNF4α—Hepatocyte nuclear factor 4 alpha; PKD—polycystic kidney disease; PPAR-α—Peroxisome proliferator-activated receptor alpha, cAMP—cyclic adenosine monophosphate; PGC1α—Peroxisome Proliferator-Activated Receptor-Gamma Coactivator 1 Alpha).

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
