# Peer review of "Metabolic Changes in Polycystic Kidney Disease as a Potential Target for Systemic Treatment"

_ijms, 2020, doi:10.3390/ijms21176093_

Round 1

Reviewer 1 Report

The paper offers a clear overview of metabolic alterations in PKD that the authors comment exhaustively. I express a positive opinion on the publication of the manuscript, provided that the following changes are made:

1) As previously done (row 54), it would be preferable, for clarity, to indicate in row 67 the protein encoded by the PKD gene (polycystin).

2) In the chapter "Mammalian target of rapamycin (mTOR) activation in PKD" only examples of ADPKD are reported, it would be appropriate to refer and comment also the role of mTOR in ARPKD (Fischer DC, Jacoby U, Pape L, et al. Activation of the AKT / mTOR pathway in autosomal recessive polycystic kidney disease (ARPKD). Nephrol Dial Transplant. 2009; 24 (6): 1819-1827; Ren XS, Sato Y, Harada K, et al. Activation of the PI3K / mTOR pathway is involved in cystic proliferation of cholangiocytes of the PCK rat. PLoS One. 2014; 9 (1): e87660. Published 2014 Jan 30).

3) References 59-60-61-62 present in the bibliography seem to have no correspondence in the text, probably in line 125 there are the wrong references.

4) Line 165: it would be correct to write “mitochondrial matrix”

5) Fenofibrate is agonist of PPAR-α (alpha), as correctly reported in figure 2  but erroneously reported as PPAR-γ in the text and caption of figure 2

6) Some grammatical / spelling errors: lines 204-217-222 etc.

Reviewer 2 Report

  1. The authors should be more careful when drafting and submitting a paper. I think the name of the last author (affiliated to 5) has been omitted. Also, the authors do not use the same font throughout the text (e.g. abbreviations). Is this a review or a mini-review?
  2. Since this is a narrative review, the authors should check whether it adheres to the SANRA checklist: https://researchintegrityjournal.biomedcentral.com/articles/10.1186/s41073-019-0064-8
  3. The figures should be presented in color to attract readers. Also, the quality of the first figure is pretty bad.
  4. The authors should also present their opinions and commentaries regarding the reviewed findings. How can we translate the content of this paper into the everyday clinical practice in the nephrology ward?
  5. Oxidative stress is an important mechanism in several kidney disorders. The authors have only briefly mentioned about the association between obesity, diabetes and oxidative stress. Please see: https://www.wjgnet.com/1948-9358/articlehighlights/v11/i5/193.htm

Reviewer 3 Report

This review describes metabolic changes in PKD for further clinical and treatment purposes. In my opinion this manuscript is of interest, very well described, clearly organized and with comprehensive clinicopathological and molecular characteristics of PKD.

Minor remarks:

  • I suggest to include ARPKD and ADPKD incidence and mortality rate.
  • Please verify all abbreviations used. Gene/protein abbreviation should be also explained in the text body (2nd page, 53rd and 71st row; 5th page, 198th -199th row; 7th page, 272nd row).
  • 4th page, 174th row – please use Otto Warburg instead of O. Warburg.
  • 6th page, 227th row (fig. 1) – please use mitochondrion instead of mitochondrium.

Round 2

Reviewer 2 Report

The authors have addressed most of the comments in my previous report, as well as the suggestions of the other two reviewers. I still believe that the figures should be presented in colour, I don't think there is a supplementary fee for this. Also, their quality should be improved.

Author Response

We have now included figures in colour.